# Learning Deep Mean Field Games for Modeling Large Population Behavior

**Jiachen Yang[1], Xiaojing Ye[2], Rakshit Trivedi[1], Huan Xu[1] & Hongyuan Zha[1]**
yjiachen@gmail.com, xye@gsu.edu, rstrivedi@gatech.edu,
huan.xu@isye.gatech.edu, zha@cc.gatech.edu
[1]Georgia Institute of Technology
[2]Georgia State University

## Abstract

We consider the problem of representing collective behavior of large populations and predicting the evolution of a population distribution over a discrete state space. A discrete time mean field game (MFG) is motivated as an interpretable model founded on game theory for understanding the aggregate effect of individual actions and predicting the temporal evolution of population distributions. We achieve a synthesis of MFG and Markov decision processes (MDP) by showing that a special MFG is reducible to an MDP. This enables us to broaden the scope of mean field game theory and infer MFG models of large real-world systems via deep inverse reinforcement learning. Our method learns both the reward function and forward dynamics of an MFG from real data, and we report the first empirical test of a mean field game model of a real-world social media population.

## 1 Introduction

> Nothing takes place in the world whose meaning is not that of some maximum or minimum. (Leonhard Euler)

Major global events shaped by large populations in social media, such as the Arab Spring, the Black Lives Matter movement, and the fake news controversy during the 2016 U.S. presidential election, provide significant impetus for devising new models that account for macroscopic population behavior resulting from the aggregate decisions and actions taken by all individuals (Howard et al., 2011; Anderson & Hitlin, 2016; Silverman, 2016). Just as physical systems behave according to the principle of least action, to which Euler's statement alludes, population behavior consists of individual actions that may be optimal with respect to some objective. The increasing usage of social media in modern societies lends plausibility to this hypothesis (Perrin, 2015), since the availability of information enables individuals to plan and act based on their observations of the global population state. For example, a population's behavior directly affects the ranking of a set of trending topics on social media, represented by the global population distribution over topics, while each user's observation of this global state influences their choice of the next topic in which to participate, thereby contributing to future population behavior (Twitter, 2017). In general, this feedback may be present in any system where the distribution of a large population over a state space is observable (or partially observable) by each individual, whose behavior policy generates actions given such observations. This motivates multiple criteria for a model of population behavior that is learnable from real data:

1. The model captures the dependency between population distribution and their actions.

2. It represents observed individual behavior as optimal for some implicit reward.

3. It enables prediction of future population distribution given measurements at previous times.

We present a mean field game (MFG) approach to address the modeling and prediction criteria. Mean field games originated as a branch of game theory that provides tractable models of large agent populations, by considering the limit of $N$-player games as $N$ tends to infinity (Lasry & Lions, 2007). In this limit, an agent population is represented via their distribution over a state space, and each agent's optimal strategy is informed by a reward that is a function of the population distribution and their aggregate actions. The stochastic differential equations that characterize MFG

can be specialized to many settings: optimal production rate of exhaustible resources such as oil among many producers (Guéant et al., 2011); optimizing between conformity to popular opinion and consistency with one's initial position in opinion networks (Bauso et al., 2016); and the transition between competing technologies with economy of scale (Lachapelle et al., 2010). Representing agents as a distribution means that MFG is scalable to arbitrary population sizes, enabling it to simulate real-world phenomenon such as the Mexican wave in stadiums (Guéant et al., 2011).

As the model detailed in Section 3 will show, MFG naturally addresses the modeling criteria in our problem context while overcoming limitations of alternative predictive methods. For example, time series analysis builds predictive models from data, but these models are incapable of representing any motivation (i.e. reward) that may produce a population's behavior policy. Alternatively, methods that employ the underlying population network structure have assumed that nodes are only influenced by a local neighborhood, do not account for a global state, and may face difficulty in explaining events as the result of any implicit optimization. (Farajtabar et al., 2015; De et al., 2016). MFG is unique as a descriptive model whose solution tells us how a system naturally behaves according to its underlying optimal control policy. This observation enables us to draw a connection with the framework of Markov decision processes (MDP) and reinforcement learning (RL) (Sutton & Barto, 1998). The crucial difference from a traditional MDP viewpoint is that we frame the problem as *MFG model inference via MDP policy optimization*: we use the MFG model to describe natural system behavior by solving an associated MDP, without imposing any control on the system. MFG offers a computationally tractable framework for adapting inverse reinforcement learning (IRL) methods (Ng & Russell, 2000; Ziebart et al., 2008; Finn et al., 2016), with flexible neural networks as function approximators, to learn complex reward functions that may explain behavior of arbitrarily large populations. In the other direction, RL enables us to devise a data-driven method for solving an MFG model of a real-world system for temporal prediction. While research on the theory of MFG has progressed rapidly in recent years, with some examples of numerical simulation of synthetic toy problems, there is a conspicuous absence of scalable methods for empirical validation (Lachapelle et al., 2010; Achdou et al., 2012; Bauso et al., 2016). Therefore, while we show how MFG is well-suited for the specific problem of modeling population behavior, we also demonstrate a general data-driven approach to MFG inference via a synthesis of MFG and MDP.

Our main contributions are the following. We propose a data-driven approach to learn an MFG model along with its reward function, showing that research in MFG need not be confined to toy problems with artificial reward functions. Specifically, we derive a discrete time graph-state MFG from general MFG and provide detailed interpretation in a real-world setting (Section 3). Then we prove that a special case can be reduced to an MDP and show that finding an optimal policy and reward function in the MDP is equivalent to inference of the MFG model (Section 4). Using our approach, we empirically validate an MFG model of a population's activity distribution on social media, achieving significantly better predictive performance compared to baselines (Section 5). Our synthesis of MFG with MDP has potential to open new research directions for both fields.

## 2 RELATED WORK

Mean field games originated in the work of Lasry & Lions (2007), and independently as stochastic dynamic games in Huang et al. (2006), both of which proposed mean field problems in the form of differential equations for modeling problems in economics and analyzed the existence and uniqueness of solutions. Guéant et al. (2011) provided a survey of MFG models and discussed various applications in continuous time and space, such as a model of population distribution that informed the choice of application in our work. Even though the MFG framework is agnostic towards the choice of cost function (i.e. negative reward), prior work make strong assumptions on the cost in order to attain analytic solutions. We take a view that the dynamics of any game is heavily impacted by the reward function, and hence we propose methods to learn the MFG reward function from data.

Discretization of MFGs in time and space have been proposed (Gomes et al., 2010; Achdou et al., 2012; Guéant, 2015), serving as the starting point for our model of population distribution over discrete topics; while these early work analyze solution properties and lack empirical verification, we focus on algorithms for attaining solutions in real-world settings. Related to our application case, prior work by Bauso et al. (2016) analyzed the evolution of opinion dynamics in multi-population environments, but they imposed a Gaussian density assumption on the initial population distribution

and restrictions on agent actions, both of which limit the generality of the model and are not assumed in our work. There is a collection of work on numerical finite-difference methods for solving continuous mean field games (Achdou et al., 2012; Lachapelle et al., 2010; Carlini & Silva, 2014). These methods involve forward-backward or Newton iterations that are sensitive to initialization and have inherent computational challenges for large real-valued state and action spaces, which limit these methods to toy problems and cannot be scaled to real-world problems. We overcome these limitations by showing how the MFG framework enables adaptation of RL algorithms that have been successful for problems involving unknown reward functions in large real-world domains.

In reinforcement learning, there are numerous value- and policy-based algorithms employing deep neural networks as function approximators for solving MDPs with large state and action spaces (Mnih et al., 2013; Silver et al., 2014; Lillicrap et al., 2015). Even though there are generalizations to multi-agent settings (Hu et al., 1998; Littman, 2001; Lowe et al., 2017), the MDP and Markov game frameworks do not easily suggest how to represent systems involving thousands of interacting agents whose actions induce an optimal trajectory through time. In our work, mean field game theory is the key to framing the modeling problem such that RL can be applied.

Methods in unknown MDP estimation and inverse reinforcement learning aim to learn an optimal policy while estimating an unknown quantity of the MDP, such as the transition law (Burnetas & Katehakis, 1997), secondary parameters (Budhiraja et al., 2012), and the reward function (Ng & Russell, 2000). The maximum entropy IRL framework has proved successful at learning reward functions from expert demonstrations (Ziebart et al., 2008; Boularias et al., 2011; Kalakrishnan et al., 2013). This probabilistic framework can be augmented with deep neural networks for learning complex reward functions from demonstration samples (Wulfmeier et al., 2015; Finn et al., 2016). Our MFG model enables us to extend the sample-based IRL algorithm in Finn et al. (2016) to the problem of learning a reward function under which a large population's behavior is optimal, and we employ a neural network to process MFG states and actions efficiently.

## 3 MEAN FIELD GAMES

We begin with an overview of a continuous-time mean field games over graphs, and derive a general discrete-time graph-state MFG (Guéant, 2015). Then we give a detailed presentation of a discrete-time MFG over a complete graph, which will be the focus for the rest of this paper.

### 3.1 MEAN FIELD GAMES ON GRAPHS

Let $\mathcal{G} = (\mathcal{V}, \mathcal{E})$ be a directed graph, where the vertex set $\mathcal{V} = \{1, \ldots, d\}$ represents $d$ possible states of each agent, and $\mathcal{E} \subseteq \mathcal{V} \times \mathcal{V}$ is the edge set consisting of all possible direct transition between states (i.e., a agent can hop from $i$ to $j$ only if $(i, j) \in \mathcal{E}$). For each node $i \in \mathcal{V}$, define $\mathcal{V}_i^+ := \{j : (j, i) \in \mathcal{E}\}$, $\mathcal{V}_i^- := \{j : (i, j) \in E\}$, and $\bar{\mathcal{V}}_i^+ := \mathcal{V}_i^+ \cup \{i\}$ and $\bar{\mathcal{V}}_i^- := \mathcal{V}_i^- \cup \{i\}$. Let $\pi_i(t)$ be the density (proportion) of agent population in state $i$ at time $t$, and $\pi(t) := (\pi_1(t), \ldots, \pi_d(t))$. Population dynamics are generated by right stochastic matrices $P(t) \in \mathbb{S}(\mathcal{G})$, where $\mathbb{S}(\mathcal{G}) := \mathbb{S}_1(\mathcal{G}) \times \cdots \times \mathbb{S}_d(\mathcal{G})$ and each row $P_i(t)$ belongs to $\mathbb{S}_i(\mathcal{G}) := \{p \in \Delta^{d-1} \mid \text{supp}(p) \subset \bar{\mathcal{V}}_i^-\}$ where $\Delta^{d-1}$ is the simplex in $\mathbb{R}^d$. Moreover, we have a value function $V_i(t)$ of state $i$ at time $t$, and a reward function $r_i(\pi(t), P_i(t))$ [1] , quantifying the instantaneous reward for agents in state $i$ taking transitions with probability $P_i(t)$ when the current distribution is $\pi(t)$. We are mainly interested in a discrete time graph state MFG, which is derived from a continuous time MFG by the following proposition. Appendix A provides a derivation from the continuous time MFG.

**Proposition 1.** *Under a semi-implicit discretization scheme with unit time step labeled by $n$, the backward Hamilton-Jacobi-Bellman (HJB) equation and the forward Fokker-Planck equation for each $i \in \{1, \ldots, d\}$ and $n = 0, \ldots, N - 1$ in a discrete time graph state MFG are given by:*

$$\text{(HJB)} \qquad V_i^n = \max_{P_i^n \in \mathbb{S}_i(\mathcal{G})} \left\{ r_i(\pi^n, P_i^n) + \sum_{j \in \bar{\mathcal{V}}_i^-} P_{ij}^n V_j^{n+1} \right\} \qquad (1)$$

$$\text{(Fokker-Planck)} \qquad \pi_i^{n+1} = \sum_{j \in \bar{\mathcal{V}}_i^+} P_{ji}^n \pi_j^n \qquad (2)$$

---

[1] We here consider a rather special formulation where the reward function $r_i$ only depends on the overall population distribution $\pi(t)$ and the choice $P_i$ the players in state $i$ made.

## 3.2 DISCRETE TIME MFG OVER COMPLETE GRAPH

Proposition 1 shows that a discrete time MFG given in Gomes et al. (2010) can be seen as a special case of a discrete time graph state MFG with a *complete* graph (such that $\mathbb{S}(\mathcal{G}) = \Delta^{d-1} \times \cdots \times \Delta^{d-1}$ ($d$ of $\Delta^{d-1}$)). We focus on the complete graph in this paper, as the methodology can be readily applied to general directed graphs. While Section 4 will show a connection between MFG and MDP, we note here that a "state" in the MFG sense is a node in $\mathcal{V}$ and not an MDP state. [2] We now interpret the model using the example of evolution of user activity distribution over topics on social media, to provide intuition and set the context for our real-world experiments in Section 5. Independent of any particular interpretation, the MFG approach is generally applicable to any problem where population size vastly outnumbers a set of discrete states.

- **Population distribution** $\pi^n \in \Delta^{d-1}$ for $n = 0, \ldots, N$. Each $\pi^n$ is a discrete probability distribution over $d$ topics, where $\pi_i^n$ is the fraction of people who posted on topic $i$ at time $n$. Although a person may participate in more than one topic within a time interval, normalization can be enforced by a small time discretization or by using a notion of "effective population size", defined as population size multiplied by the max participation count of any person during any time interval. $\pi^0$ is a given initial distribution.

- **Transition matrix** $P^n \in \mathbb{S}(\mathcal{G})$. $P_{ij}^n$ is the probability of people in topic $i$ switching to topic $j$ at time $n$, so we refer to $P_i^n$ as the *action* of people in topic $i$. $P^n$ generates the forward equation

$$\pi_j^{n+1} = \sum_{i=1}^d P_{ij}^n \pi_i^n \tag{3}$$

- **Reward** $r_i(\pi^n, P_i^n) := \sum_{j=1}^d P_{ij}^n r_{ij}(\pi^n, P_i^n)$, for $i \in \{1, \ldots, d\}$. This is the reward received by people in topic $i$ who choose action $P_i^n$ at time $n$, when the distribution is $\pi^n$. In contrast to previous work, we learn the reward function from data (Section 4.1). We make a *locality assumption*: reward for $i$ depends only on $P_i^n$, not on the entire $P^n$, which means that actions by people in $j \neq i$ have no instantaneous effect on the reward for people in topic $i$. [3]

- **Value function** $V^n \in \mathbb{R}^d$. $V_i^n$ is the expected maximum total reward of being in topic $i$ at time $n$. A terminal value $V^N$ is given, which we set to zero to avoid making any assumption on the problem structure beyond what is contained in the learned reward function.

- **Average reward** $e_i(\pi, P, V)$, for $i \in \{1, \ldots, d\}$ and $V \in \mathbb{R}^d$ and $P \in \mathbb{S}(\mathcal{G})$. This is the average reward received by agents at topic $i$ when the current distribution is $\pi$, action $P$ is chosen, and the subsequent expected maximum total reward is $V$. For a general $r_{ij}(\pi, P)$, it is defined as:

$$e_i(\pi, P, V) = \sum_{j=1}^d P_{ij}(r_{ij}(\pi, P) + V_j) \tag{4}$$

Intuitively, agents want to act optimally in order to maximize their expected total average reward. For $P \in \mathbb{S}(\mathcal{G})$ and a vector $q \in \mathbb{S}_i(\mathcal{G})$, define $\mathcal{P}(P, i, q)$ to be the matrix equal to $P$, except with the $i$-th row replaced by $q$. Then a *Nash maximizer* is defined as follows:

**Definition 1.** A right stochastic matrix $P \in \mathbb{S}(\mathcal{G})$ is a *Nash maximizer* of $e(\pi, P, V)$ if, given a fixed $\pi \in \Delta^{d-1}$ and a fixed $V \in \mathbb{R}^d$, there is

$$e_i(\pi, P, V) \geq e_i(\pi, \mathcal{P}(P, i, q), V) \tag{5}$$

for any $i \in \{1, \ldots, d\}$ and any $q \in \mathbb{S}_i(\mathcal{G})$.

The rows of $P$ form a Nash equilibrium set of actions, since for any topic $i$, the people in topic $i$ cannot increase their reward by unilaterally switching their action from $P_i$ to any $q$. Under Definition 1, the value function of each topic $i$ at each time $n$ satisfies the optimality criteria:

$$V_i^n = \max_{q \in \mathbb{S}_i(\mathcal{G})} \left\{ \sum_{j=1}^d q_j \left[ r_{ij}(\pi^n, \mathcal{P}(P^n, i, q)) + V_j^{n+1} \right] \right\} \tag{6}$$

A solution of the MFG is a sequence of pairs $\{(\pi^n, V^n)\}_{n=0,\ldots,N}$ satisfying optimality criteria (6) and forward equation (3).

---

[2] Section 4 explains that the population distribution $\pi$ is the appropriate definition of an MDP state.

[3] If this assumption is removed, there is a resemblance between the discrete time MFG and a Markov game in a continuous state and continuous action space (Littman, 2001; Hu et al., 1998). However, it turns out that the general MFG is a strict generalization of a multi-agent MDP (Appendix G).

## 4 INFERENCE OF MFG VIA MDP OPTIMIZATION

A Markov decision process is a well-known framework for optimization problems. We focus on the discrete time MFG in Section 3.2 and prove a reduction to a finite-horizon deterministic MDP, whose state trajectory under an optimal policy coincides with the forward evolution of the MFG. This leads to the essential insight that solving the *optimization* problem of an MDP is equivalent to solving an MFG that *describes* population behavior. This connection will enable us to apply efficient inverse RL methods, using measured population trajectories, to learn an MFG model along with its reward function in Section 4.1. The MDP is constructed as follows:

**Definition 2.** A finite-horizon deterministic MDP for a discrete time MFG over a complete graph is defined as:

- States: $\pi^n \in \Delta^{d-1}$, the population distribution at time $n$.

- Actions: $P^n \in \mathbb{S}(\mathcal{G})$, the transition probability matrix at time $n$.

- Reward: $R(\pi^n, P^n) := \sum_{i=1}^d \pi_i^n \sum_{j=1}^d P_{ij}^n r_{ij}(\pi^n, P_i^n)$

- Finite-horizon state transition, given by Eq (3): $\forall n \in \{0, \ldots, N-1\}\colon \pi_j^{n+1} = \sum_{i=1}^d P_{ij}^n \pi_i^n$.

**Theorem 2.** *The value function of a solution to the discrete time MFG over a complete graph defined by optimality criteria (6) and forward equation (3) is a solution to the Bellman optimality equation of the MDP in Definition 2.*

*Proof.* Since $r_{ij}$ depends on $P^n$ only through row $P_i^n$, optimality criteria 6 can be written as

$$V_i^n = \max_{P_i \in \mathbb{S}_i(\mathcal{G})} \left\{ \sum_j P_{ij} r_{ij}(\pi^n, P_i) + \sum_j P_{ij} V_j^{n+1} \right\}. \tag{7}$$

We now define $V^*(\pi^n)$ as follows and show that it is the value function of the constructed MDP in Definition 2 by verifying that it satisfies the Bellman optimality equation:

$$V^*(\pi^n) := \sum_{i=1}^d \pi_i^n V_i^n = \sum_{i=1}^d \pi_i^n \max_{P_i \in \mathbb{S}_i(\mathcal{G})} \left\{ \sum_{j=1}^d P_{ij} r_{ij}(\pi^n, P_i) + \sum_{j=1}^d P_{ij} V_j^{n+1} \right\} \tag{8}$$

$$= \max_{P \in \mathbb{S}(\mathcal{G})} \left\{ \sum_{i=1}^d \pi_i^n \sum_{j=1}^d P_{ij} r_{ij}(\pi^n, P_i) + \sum_{j=1}^d \left( \sum_{i=1}^d P_{ij} \pi_i^n \right) V_j^{n+1} \right\} \tag{9}$$

$$= \max_{P \in \mathbb{S}(\mathcal{G})} \left\{ R(\pi^n, P) + \sum_{j=1}^d \pi_j^{n+1} V_j^{n+1} \right\} \tag{10}$$

$$= \max_{P \in \mathbb{S}(\mathcal{G})} \left\{ R(\pi^n, P) + V^*(\pi^{n+1}) \right\} \tag{11}$$

which is the Bellman optimality equation for the MDP in Definition 2. $\qquad \square$

**Corollary 1.** *Given a start state $\pi^0$, the state trajectory under the optimal policy of the MDP in Definition 2 is equivalent to the forward evolution part of the solution to the MFG.*

*Proof.* Under the optimal policy, equations 11 and 8 are satisfied, which means the matrix $P$ generated by the optimal policy at any state $\pi^n$ is the Nash maximizer matrix. Therefore, the state trajectory $\{\pi^n\}_{n=0,\ldots,N}$ is the forward part of the MFG solution. $\qquad \square$

### 4.1 REINFORCEMENT LEARNING SOLUTION FOR MFG

MFG provides a general framework for addressing the problem of modeling population dynamics, while the new connection between MFG and MDP enables us to apply inverse RL algorithms to solve the MDP in Definition 2 with unknown reward. In contrast to previous MFG research, most of which impose reward functions that are quadratic in actions and logarithmic in the state distribution

(Guéant, 2009; Lachapelle et al., 2010; Bauso et al., 2016), we learn a reward function using demonstration trajectories measured from actual population behavior, to ground the MFG representation of population dynamics on real data.

We leverage the MFG forward dynamics (Eq 3) in a sample-based IRL method based on the maximum entropy IRL framework (Ziebart et al., 2008). From this probabilistic viewpoint, we minimize the relative entropy between a probability distribution $p(\tau)$ over a space of trajectories $T := \{\tau_i\}_i$ and a distribution $q(\tau)$ from which demonstrated expert trajectories are generated (Boularias et al., 2011). This is related to a path integral IRL formulation, where the likelihood of measured optimal trajectories is evaluated only using trajectories generated from their local neighborhood, rather than uniformly over the whole trajectory space (Kalakrishnan et al., 2013). Specifically, making no assumption on the true distribution of optimal demonstration other than matching of reward expectation, we posit that demonstration trajectories $\tau_i = (\pi^0, P^1, \ldots, \pi^{N-1}, P^{N-1})_i$ are sampled from the maximum entropy distribution (Jaynes, 1957):

$$p(\tau) = \frac{1}{Z} \exp(R_W(\tau)) \tag{12}$$

where $R_W(\tau) = \sum_n R_W(\pi^n, P^n)$ is the sum of reward of single state-action pairs over a trajectory $\tau$, and $W$ are the parameters of the reward function approximator (derivation in Appendix E). Intuitively, this means that trajectories with higher reward are exponentially more likely to be sampled. Given $M$ sample trajectories $\tau_j \in \mathcal{D}_{\text{samp}}$ from $k$ distributions $F_1(\tau), \ldots, F_k(\tau)$, an unbiased estimator of the partition function $Z = \int \exp(R_W(\tau)) d\tau$ using multiple importance sampling is $\hat{Z} := \frac{1}{M} \sum_{\tau_j} z_j \exp(R_W(\tau_j))$ (Owen & Zhou, 2000), where importance weights are $z_j := \left[\frac{1}{k} \sum_k F_k(\tau_j)\right]^{-1}$ (derivation in Appendix F). Each action matrix $P$ is sampled from a stochastic policy $F_k(P; \pi, \theta)$ (overloading notation with $F(\tau)$), where $\pi$ is the current state and $\theta$ the policy parameter. The negative log likelihood of $L$ demonstration trajectories $\tau_i \in \mathcal{D}_{\text{demo}}$ is:

$$\mathcal{L}(W) = -\frac{1}{L} \sum_{\tau_i \in \mathcal{D}_{\text{demo}}} R_W(\tau_i) + \log\left(\frac{1}{M} \sum_{\tau_j \in \mathcal{D}_{\text{samp}}} z_j \exp(R_W(\tau_j))\right) \tag{13}$$

We build on Guided Cost Learning (GCL) in Finn et al. (2016) (Alg 1) to learn a deep neural network approximation of $R_W(\pi, P)$ via stochastic gradient descent on $\mathcal{L}(W)$, and learn a policy $F(P; \pi, \theta)$ using a simple actor-critic algorithm (Sutton & Barto, 1998). In contrast to GCL, we employ a combination of convolutional neural nets and fully-connected layers to process both the action matrix $P$ and state vector $\pi$ efficiently in a single architecture (Appendix C), analogous to how Lillicrap et al. (2015) handle image states in Atari games. Due to our choice of policy parameterization (described below), we also set importance weights to unity for numerical stability. These implementation choices result in successful learning of a reward representation (Fig 1).

Our forward MDP solver (Alg 2) performs gradient ascent on the policy's expected start value $\mathbb{E}[v(\pi^0)|F(P; \pi, \theta)]$ w.r.t. $\theta$, to find successively better policies $F_k(P; \pi, \theta)$. We construct the joint distribution $F(P; \pi, \theta)$ informed by domain knowledge about human population behavior on social media, but this does not reduce the generality of the MFG framework since it is straightforward to employ flexible policy and value networks in a DDPG algorithm when intuition is not available (Silver et al., 2014; Lillicrap et al., 2015). Our joint distribution is $d$ instances of a $d$-dimensional Dirichlet distribution, each parameterized by an $\alpha^i \in \mathbb{R}_+^d$. Each row $P_i$ is sampled from

$$f(P_{i1}, \ldots, P_{id}; \alpha_1^i, \ldots, \alpha_d^i) = \frac{1}{B(\alpha^i)} \prod_{j=1}^{d} (P_{ij})^{\alpha_j^i - 1} \tag{14}$$

where $B(\cdot)$ is the Beta function and $\alpha_j^i$ is defined using the softplus function $\alpha_j^i(\pi, \theta) := \ln(1 + \exp\{\theta(\pi_j - \pi_i)\})$, which is a monotonically increasing function of the population density difference $\pi_j - \pi_i$. In practice, a constant scaling factor $c \in \mathbb{R}$ can be applied to $\alpha$ for variance reduction. Finally, we let $F(P^n; \pi^n, \theta) = \prod_{i=1}^{d} f(P_i^n; \alpha^i(\pi^n, \theta))$ denote the parameterized policy, from which $P^n$ is sampled based on $\pi^n$, and whose logarithmic gradient $\nabla_\theta \ln(F)$ can be used in a policy gradient algorithm. We learned an approximate value function $\hat{V}(\pi; w)$ as a baseline for variance reduction, approximated as a linear combination of all polynomial features of $\pi$ up to second order, with parameter $w$ (Sutton et al., 2000).

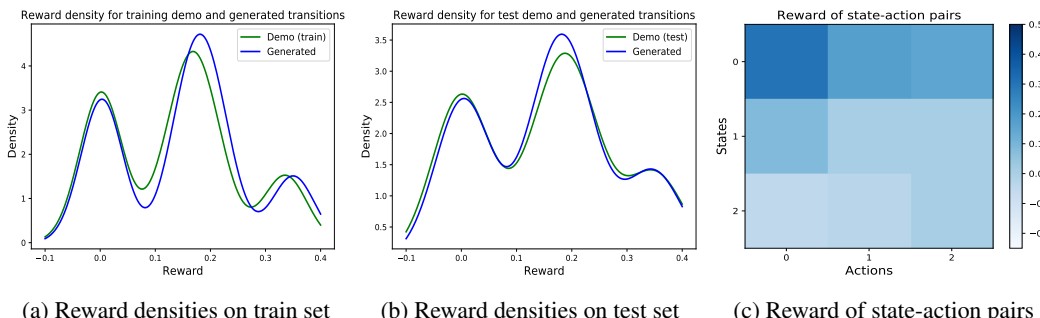

(a) Reward densities on train set       (b) Reward densities on test set       (c) Reward of state-action pairs

Figure 1: (a) JSD between train demo and generated transitions is 0.130. (b) JSD between test demo and generated transitions is 0.017. (c) Reward of state-action pairs. States: large negative mass gradient from $\pi_1$ to $\pi_d$ (S0), less negative gradient (S1), uniform (S2). Actions: high probability transitions to smaller indices (A0), uniform transition (A1), row-reverse of A0 (A2).

## 5 EXPERIMENTS

We demonstrate the effectiveness of our method with two sets of experiments: (i) inference of an interpretable reward function and (ii) prediction of population trajectory over time. Our experiment matches the discrete time mean field game given in Section 3.2: we use data representing the activity of a Twitter population consisting of 406 users. We model the evolution of the population distribution over $d = 15$ topics and $N = 16$ time steps (9am to midnight) each day for 27 days. The sequence of state-action pairs $\{(\pi^n, P^n)\}_{n=0,...,N-1}$ measured on each day shall be called a *demonstration trajectory*. Although the set of topics differ semantically each day, indexing topics in order of decreasing initial popularity suffices for identifying the topic sets across all days. As explained earlier, the MFG framework can model populations of arbitrarily large size, and we find that our chosen size is sufficient for extracting an informative reward and policy from the data. For evaluating performance on trajectory prediction, we compare MFG with two baselines:
**VAR.** Vector autoregression of order 18 trained on 21 demonstration trajectories.
**RNN.** Recurrent neural network with a single fully-connected layer and rectifier nonlinearity.
We use Jenson-Shanon Divergence (JSD) as metric to report all our results. Appendix D provides comprehensive implementation details.

### 5.1 INTERPRETATION OF REWARD FUNCTION

We evaluated the reward using four sets of state-action pairs acquired from: 1. all train demo trajectories; 2. trajectories generated by the learned policy given initial states $\pi^0$ of train trajectories; 3. all test demo trajectories; 4. trajectories generated by the learned policy given initial states $\pi^0$ of test trajectories. We find three distinct modes in the density of reward values for both the train group of sets 1 and 2 (Fig 1a) and the test group of sets 3 and 4 (Fig 1b). Although we do not have access to a ground truth reward function, the low JSD values of 0.13 and 0.017 between reward distributions for demo and generated state-action pairs show generalizability of the learned reward function. We further investigated the reward landscape with nine state-action pairs (Figure 1c), and find that the mode with highest rewards is attained by pairing states that have large mass in topics having high initial popularity (S0) with action matrices that favor transition to topics with higher density (A0). Uniformly distributed state vectors (S2) attain the lowest rewards, and states with a small negative mass gradient from topic 1 to topic $d$ (S1) attain medium rewards. Simply put, MFG agents who optimize for this reward are more likely to move towards more popular topics. While this numerical exploration of the reward reveals interpretable patterns, the connection between such rewards learned via our method and any optimization process in the population requires more empirical study.

### 5.2 TRAJECTORY PREDICTION

To test the usefulness of the reward and MFG model for prediction, the learned policy was used with the forward equation to generate complete trajectories, given initial distributions. Fig 2a (log scale) shows that MFG has $58\%$ smaller error than VAR when evaluated on the JSD between generated

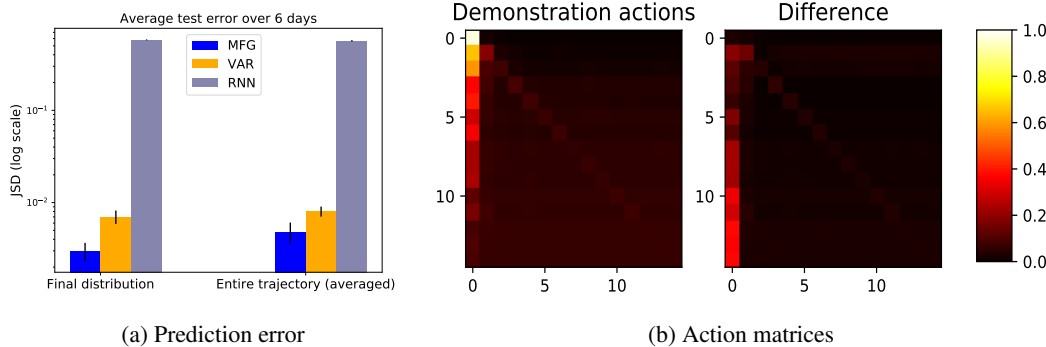

(a) Prediction error

(b) Action matrices

Figure 2: (a) Test error on final distribution and mean over entire trajectory (log scale). MFG: (2.9e-3, 4.9e-3), VAR: (7.0e-3, 8.1e-3), RNN: (0.58, 0.57). (b) heatmap of action matrix $P \in \mathbb{R}^{15 \times 15}$ averaged element-wise over demo train set, and absolute difference between average demo action matrix and average matrix generated from learned policy.

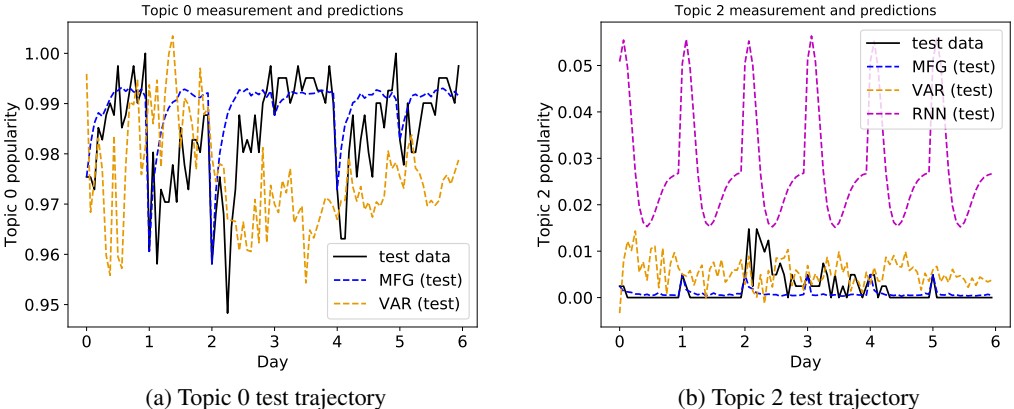

(a) Topic 0 test trajectory

(b) Topic 2 test trajectory

Figure 3: (a) Measured and predicted trajectory of topic 0 popularity over test days for MFG and VAR (RNN outside range and not shown). (b) Measured and predicted trajectory of topic 2 popularity over test days for all methods.

and measured *final* distributions $\mathrm{JSD}(\pi_{\mathrm{generated}}^{N-1}, \pi_{\mathrm{measured}}^{N-1})$, and $40\%$ smaller error when evaluated on the average JSD over all hours in a day $\frac{1}{N} \sum_{n=0}^{N-1} \mathrm{JSD}(\pi_{\mathrm{generated}}^{n}, \pi_{\mathrm{measured}}^{n})$. Both measures were averaged over $M = 6$ held-out test trajectories. It is worth emphasizing that learning the MFG model required *only the initial population distribution* of each day in the training set (line 4 in Alg 2), while VAR and RNN used the distributions over all hours of each day. MFG achieves better prediction performance even with fewer training samples, possibly because it is a more structured approximation of the true mechanism underlying population dynamics, in contrast to VAR and RNN that rely on regression. As shown by sample trajectories for topic 0 and 2 in Figures 3, and the average transition matrices in Figure 2b, MFG correctly represents the fact that the real population tends to congregate to topics with higher initial popularity (lower topic indices), and that the popularity of topic 0 becomes more dominant across time in each day. The small real-world dataset size, and the fact that RNN mainly learns state transitions without accounting for actions, could be contributing factors to the lower performance of RNN. We acknowledge that our design of policy parameterization, although informed by domain knowledge, introduced bias and resulted in noticeable differences between demonstration and generated transition matrices. This can be addressed using deep policy and value networks, since the MFG framework is agnostic towards choice of policy representation.

## 6 CONCLUSION

We have motivated and demonstrated a data-driven method to solve a mean field game model of population evolution, by proving a connection to Markov decision processes and building on methods

in reinforcement learning. Our method is scalable to arbitrarily large populations, because the MFG framework represents population density rather than individual agents, while the representations are linear in the number of MFG states and quadratic in the transition matrix. Our experiments on real data show that MFG is a powerful framework for learning a reward and policy that can predict trajectories of a real world population more accurately than alternatives. Even with a simple policy parameterization designed via some domain knowledge, our method attained superior performance on test data. It motivates exploration of flexible neural networks for more complex applications.

An interesting extension is to develop an efficient method for solving the discrete time MFG in a more general setting, where the reward at each state $i$ is coupled to the full population transition matrix. Our work also opens the path to a variety of real-world applications, such as a synthesis of MFG with models of social networks at the level of individual connections to construct a more complete model of social dynamics, and mean field models of interdependent systems that may display complex interactions via coupling through global states and reward functions.

### ACKNOWLEDGMENTS

We sincerely thank our anonymous ICLR reviewers for critical feedback that helped us to improve the clarity and precision of our presentation. This work was supported in part by NSF CMMI-1745382 and NSF IIS-1717916.

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

# A    PROOF OF PROPOSTION 1

Given the definitions in Section 3.1, a mean field game is defined by a Hamilton-Jacobi-Bellman (HJB) equation evolving backwards in time and a Fokker-Planck equation evolving forward in time. The continuous-time Hamilton-Jacobi-Bellman (HJB) equation on $\mathcal{G}$ is

$$V_i'(t) = -\max_{P_i} \left\{ \sum_{j \in \bar{\mathcal{V}}_i^-} P_{ij}(t)(V_j(t) - V_i(t)) + r_i(\pi(t), P_i(t)) \right\} \tag{15}$$

where $r_i(\pi, P_i)$ is the reward function, and $V_i(t)$ is the value function of state $i$ at time $t$. Note that the reward function $r_i(\pi(t), P_i(t))$ is often presented as $-c_i(\pi(t), P_i(t))$ for some cost function $c_i(\pi(t), P_i(t))$ in the MFG context, and similarly for $V_i(t)$. In addition, we set $r_i(\pi(t), P_i(t)) = -\infty$ if $P_i(t) \notin \mathbb{S}_i(\mathcal{G})$ (i.e. $P(t)$ must be a valid transition matrix). For any fixed $\pi(t)$, let $\mathcal{H}_i(\pi(t), \cdot)$ be the Legendre transform of $c_i(\pi(t), \cdot)$ defined by

$$\mathcal{H}_i(\pi(t), \cdot) = \max_{P_i} \{\langle \cdot, P_i \rangle - c_i(\pi(t), P_i)\} = \max_{P_i} \{\langle \cdot, P_i \rangle + r_i(\pi(t), P_i)\} \tag{16}$$

Then the HJB equation (15) is an analogue to the *backward equation* in mean field games

$$V_i'(t) + \mathcal{H}_i(\pi(t), [V_j(t) - V_i(t)]_{j \in \bar{\mathcal{V}}_i^-}) = 0 \tag{17}$$

where $[V_j(t) - V_i(t)]_{j \in \bar{\mathcal{V}}_i^-} \in \mathbb{R}^{|\bar{\mathcal{V}}_i^-|}$ is the dual variable of $P_i$. We can discretize (15) using a semi-implicit scheme with unit time step labeled by $n$ to obtain

$$V_i^{n+1} - V_i^n = -\max_{P_i} \left\{ \sum_{j \in \bar{\mathcal{V}}_i^-} P_{ij}^n(V_j^{n+1} - V_i^{n+1}) + r_i(\pi^n, P_i^n) \right\} \tag{18}$$

Rearranging (18) yields the discrete time HJB equation over a graph (19)

$$V_i^n = \max_{P_i} \left\{ r_i(\pi^n, P_i^n) + \sum_{j \in \bar{\mathcal{V}}_i^-} P_{ij}^n V_j^{n+1} \right\} \tag{19}$$

The *forward* evolving Fokker-Planck equation for the continuous-time graph-state MFG is given by

$$\pi_i'(t) = \sum_{j \in \mathcal{V}_i^+} Q_{ji}(t)\pi_j(t) - \sum_{j \in \mathcal{V}_i^-} Q_{ij}(t)\pi_i(t) \tag{20}$$

$$\text{where} \quad Q_{ji}(t) = \partial_{u_i} \mathcal{H}_j(\pi(t), [V_k(t) - V_j(t)]_{k \in \bar{\mathcal{V}}_j^-}) \tag{21}$$

where $\partial_{u_i} \mathcal{H}_j(\pi, u)$ is the partial derivative w.r.t. the coordinate corresponding to the $i$-th index of the argument $u \in \mathbb{R}^{|\bar{\mathcal{V}}_j^-|}$. We can set $Q_{ji}(t) = 0$ for all $(j, i) \notin \mathcal{E}$, so that $Q(t) := [Q_{ji}(t)]$ can be regarded as the $d$-by-$d$ infinitesimal generator matrix of states $\pi(t)$, and hence (20) can be written as $\pi'(t) = \pi(t)Q(t)$, where $\pi(t) \in \mathbb{R}^d$ is a row vector. Then an Euler discretization of (20) with unit time step reduces to $\pi^{n+1} - \pi^n = \pi^n Q^n$, which can be written as

$$\pi_i^{n+1} = \sum_{j \in \bar{\mathcal{V}}_i^+} P_{ji}^n \pi_j^n \tag{22}$$

where $P_{ij}^n := Q_{ij}^n + \delta_{ij}$. If the graph $\mathcal{G}$ is complete, meaning $\mathcal{E} = \{(i, j) : 1 \leq i, j \leq d\}$, then the summation is taken over $j = 1, \ldots, d$. For ease of presentation, we only consider the complete graph in this paper, as all derivations can be carried out similarly for general directed graphs. A solution of a mean field game defined by (19) and (22) is a collection of $V_i^n$ and $\pi_i^n$ for $i = 1, \ldots, d$ and $n = 0, \ldots, N$.

## B  ALGORITHMS

We learn a reward function and policy using an adaptation of GCL (Finn et al., 2016) in Alg 1 and a simple actor-critic Alg 2 (Sutton & Barto, 1998) as a forward RL solver.

---

**Algorithm 1** Guided cost learning

---

1: **procedure** GUIDED COST LEARNING
2:     Initialize $F_0(P; \pi, \theta)$ as random policy and reward network weights $W^0$
3:     **for** iteration 1 to $I$ **do**
4:         Generate sample trajectories $\mathcal{D}_{\text{traj}}$ from $F_k(P; \pi, \theta)$
5:         $\mathcal{D}_{\text{samp}} \leftarrow \mathcal{D}_{\text{samp}} \cup \mathcal{D}_{\text{traj}}$
6:         **while** $\left| \text{Avg}_{\pi_i, P_i \sim \mathcal{D}_{\text{demo}}} (R_{W^t}(\pi_i, P_i) - R_{W^{t-1}}(\pi_i, P_i)) \right| > dR$ **do**
7:             Sample demonstration $\hat{\mathcal{D}}_{\text{demo}} \subset \mathcal{D}_{\text{demo}}$ from expert demonstration
8:             Sample $\hat{\mathcal{D}}_{\text{samp}} \subset \mathcal{D}_{\text{samp}}$
9:             $W^{t+1} \leftarrow W^t - \epsilon \nabla \mathcal{L}(W^t)$ using $\hat{\mathcal{D}}_{\text{demo}}$ and $\hat{\mathcal{D}}_{\text{samp}}$
10:         **end while**
11:         Run Alg 2 with new $R_W$ for improved $F_{k+1}$
12:     **end for**
13:     **return** Final reward function $R_W(\pi, P)$ and policy $F(P; \pi, \theta)$
14: **end procedure**

---

---

**Algorithm 2** Actor-critic algorithm for MFG

---

**Input:** Generative model $F(P; \pi, \theta)$, value function $\hat{V}(\pi; w)$, training data $\{\pi^0\}_{\text{M days}}$
**Output:** Policy parameter $\theta$, value function parameter $w$
1: **procedure** ACTOR-CRITIC-MFG($F, \hat{V}, \{\pi^0\}_{\text{M days}}, \beta, \xi, R_W$)
2:     initialize $\theta$ and $w$
3:     **for** episodes $s = 1, \ldots, S$ **do**
4:         Sample initial distribution $\pi^0$ from $\{\pi^0\}_{\text{M days}}$
5:         **for** time step $n = 0, \ldots, N - 1$ **do**
6:             Sample action $P^n \sim F(P; \pi^n, \theta)$
7:             Generate $\pi^{n+1}$ using Eq 3
8:             Receive reward $R_W(\pi^n, P^n)$
9:             $\delta \leftarrow R + \hat{V}(\pi^{n+1}; w) - \hat{V}(\pi^n; w)$
10:             $w \leftarrow w + \xi \delta \nabla_w \hat{V}(\pi^n; w)$
11:             $\theta \leftarrow \theta + \beta \delta \nabla_\theta \log(F(P; \pi^n, \theta))$
12:         **end for**
13:     **end for**
14: **end procedure**

---

## C  REWARD NETWORK

Our reward network uses two convolutional layers to process the $15 \times 15$ action matrix $P$, which is then flattened and concatenated with the state vector $\pi$ and processed by two fully-connected layers regularized with L1 and L2 penalties and dropout (probability 0.6). The first convolutional layer zero-pads the input into a $19 \times 19$ matrix and convolves one filter of kernel size $5 \times 5$ with stride 1 and applies a rectifier nonlinearity. The second convolutional layer zero-pads its input into a $17 \times 17$ matrix and convolves 2 filters of kernel size $3 \times 3$ with stride 1 and applies a rectifier nonlinearity. The fully connected layers have 8 and 4 hidden rectifier units respectively, and the output is a single fully connected tanh unit. All layers were initialized using the Xavier normal initializer in Tensorflow.

## D    EXPERIMENT DETAILS

By default, Twitter users in a certain geographical region primarily see the trending topics specific to that region (Twitter, 2017). This experiment focused on the population and trending topics in the city of Atlanta in the U.S. state of Georgia. First, a set of 406 active users were collected to form the fixed population. This was done by collecting a set of high-visibility accounts in Atlanta (e.g. the Atlanta Falcons team), gathering all Twitter users who follow these accounts, filtering for those whose location was set to Atlanta, and filtering for those who responded to least two trending topics within four days.

Data collection proceeded as follows for 27 days: at 9am of each day, a list of the top 14 trending topics on Twitter in Atlanta was recorded; for each hour until midnight, for each topic, the number of users who responded to the topic and the transition counts among topics within the past hour was recorded. Whether or not a user responded to a topic was determined by checking for posts by the user containing unique words for that topic; the "hashtag" convention of trending topics on Twitter reduces the likelihood of false positives. The hourly count of people who did not respond to any topic was recorded as the count for a "null topic". Although some users may respond to more than one topic within each hour, the data shows that this is negligible, and a shorter time interval can be used to reduce this effect. The result of data collection is a set of trajectories, one trajectory per day, where each trajectory consists of hourly measurements of the population distribution over $d = 15$ topics and their transition matrix over $N = 16$ hours.

The training set consists of trajectories $\{\pi^{0,m}, P^{0,m}, \ldots, P^{N-2,m}, \pi^{N-1,m}\}_{m=1,\ldots,M}$ over the first $M = 21$ days. MFG uses the initial distribution $\pi^0$ of each day, along with the transition equation of the constructed MDP and the policy $F(P; \pi^n, \theta)$, to produce complete trajectories for training (Alg 2 lines 4,6,7). In contrast, VAR and RNN are supervised learning methods and they use all measured distributions. RNN employs a simple recurrent unit with ReLU as nonlinear activation and weight matrix of dimension $d \times d$. VAR was implemented using the Statsmodels module in Python, with order 18 selected via random sub-sampling validation with validation set size 5 (Seabold & Perktold, 2010). For prediction accuracy, all three methods were evaluated against data from 6 held-out test days. Table 1 shows parameters of Alg 2 and 1.

Table 1: Parameters

| Parameter | Use | Value |
|---|---|---|
| $S$ | max actor-critic episodes | 4000 |
| $\beta$ | critic learning rate | $O(1/s)$ |
| $\xi$ | actor learning rate | $O(1/s \ln \ln s)$ |
| $c$ | $\alpha^i_j$ scaling factor | 1e4 |
| $\epsilon$ | Adam optimizer learning rate for reward | 1e-4 |
| $dR$ | convergence threshold for reward iteration | 1e-4 |
| $\theta_{\text{final}}$ | learned policy parameter | 8.64 |

## E    MAXIMUM ENTROPY DISTRIBUTION

Given a finite set of trajectories $\{\tau_i\}_i$, where each trajectory is a sequence of state-action pairs $\tau_i = (s_{i1}, a_{i1}, \ldots,)$. Suppose each trajectory $\tau_i$ has an unknown probability $p_i$. The entropy of the probability distribution is $H = -\sum_i p_i \ln(p_i)$. In the continuous case, we write the differential entropy:

$$H = -\int p(\tau) \ln(p(\tau)) d\tau$$

where $p(\cdot)$ is the probability density we want to derive. The constraints are:

$$\int r(\tau) p(\tau) d\tau = \mathbb{E}[r(\tau)] = \mu_r$$

$$\int p(\tau) d\tau = 1$$

The first constraint says: the expected reward over all trajectories is equal to an empirical measurement $\mu_r$. We write the Lagrangian $\mathcal{L}$:

$$\mathcal{L} = -\int p(\tau)\ln(p(\tau))d\tau - \lambda_1\left(\int p(\tau)d\tau - 1\right) - \lambda_2\left(\int r(\tau)p(\tau)d\tau - \mu_r\right)$$

For $\mathcal{L}$ to be stationary, the Euler-Lagrange equation with integrand denoted by $L$ says

$$\frac{\partial L}{\partial p} = 0$$

since $L$ does not depend on $\frac{dp}{d\tau}$. Hence

$$\lambda_1 = \ln\left(\int e^{-\lambda_2 r(\tau)}d\tau\right) - 1$$

$$p(\tau) = \exp\left(-\ln\left(\int e^{-\lambda_2 r(\tau)}d\tau\right) - \lambda_2 r(\tau)\right) = \frac{1}{Z(\lambda_2)}e^{-\lambda_2 r(\tau)}$$

where $Z := \int e^{-\lambda_2 r(\tau)}d\tau$. Then the constant $\lambda_2$ is determined by:

$$\mu_r = \int p(\tau)r(\tau)d\tau = \frac{1}{Z(\lambda_2)}\int e^{-\lambda_2 r(\tau)}r(\tau)d\tau$$

$$= -\frac{\partial}{\partial\lambda_2}\ln(Z(\lambda_2))$$

## F  MULTIPLE IMPORTANCE SAMPLING

We show how multiple importance sampling (Owen & Zhou, 2000) can be used to estimate the partition function in the maximum entropy IRL framework. The problem is to estimate $Z := \int f(x)dx$. Let $p_1, \ldots, p_m$ be $m$ proposal distributions, with $n_j$ samples from the $j$-th proposal distribution, so that samples can be denoted $X_{ij}$ for $i = 1, \ldots, n_j$ and $j = 1, \ldots, m$. Let $w_j(x)$ for $j = 1, \ldots, m$ satisfy

$$0 \leq w_j(x) \leq \sum_{j=1}^{m} w_j(x) = 1$$

Then define the estimator

$$\hat{Z} = \sum_{j=1}^{m}\frac{1}{n_j}\sum_{i=1}^{n_j} w_j(X_{ij})\frac{f(X_{ij})}{p_j(X_{ij})}$$

Let $S(p_j) = \{x \mid p_j(x) > 0\}$ be the support of $p_j$ and $S(w_j) = \{x \mid w_j(x) > 0\}$ be the support of $w_j$, and let them satisfy $S(w_j) \subset S(p_j)$. Under these assumptions:

$$\mathbb{E}[\hat{Z}] = \int f(x)dx = Z$$

In particular, choose

$$w_j(x) := \frac{n_j p_j(x)}{\sum_{k=1}^{m} n_k p_k(x)}$$

Then the estimate becomes

$$\hat{Z} = \sum_{j=1}^{m}\sum_{i=1}^{n_j}\frac{f(X_{ji})}{\sum_{k=1}^{m} n_k p_k(x)}$$

$$= \frac{1}{n}\sum_{j=1}^{m}\sum_{i=1}^{n_j}\frac{f(X_{ji})}{\sum_{k=1}^{m}\frac{n_k}{n} p_k(x)}$$

where $n = \sum_{j=1}^{m} n_j$ is the total count of samples. Further assuming that samples are drawn uniformly from all proposal distributions, so that $n_j = n_k = n/m$ for all $j, k \in \{1, \ldots, m\}$, the expression for $\hat{Z}$ reduces to the form used in Eq 13:

$$\hat{Z} = \frac{1}{n}\sum_{\text{all samples}}\frac{f(x)}{\frac{1}{m}\sum_{k=1}^{m} p_k(x)}$$

## G  A COMPARISON OF MEAN FIELD GAMES AND MULTI-AGENT MDPS

In this section, we discuss the reason that the general MFG, whose reward function $r_{ij}(\pi^n, P^n)$ depends on the full Nash maximizer matrix $P^n$, is neither reducible to a collection of distinct single-agent MDPs nor equivalent to a multi-agent MDP. Let a state in the discete space MFG be called a "topic", to avoid confounding with an MDP state.

### G.1  COLLECTION OF SINGLE-AGENT MDPS

Consider each topic $i$ as a separate entity associated with a value, rather than subsuming it into an average (as is the case in Section 4). In order to assign a value to each topic, each tuple $(i, \pi^n)$ must be defined as a state, which leads to the problem: since a state requires specification of $\pi^n$, and state transitions depend on the actions for all other topics, the action at each topic is not sufficient for fully specifying the next state. More formally, consider a value function on the state:

$$V(i, \pi^n) = \max_{q \in \mathbb{S}_i(\mathcal{G})} \left\{ \sum_j q_j r_{ij}(\pi^n, \mathcal{P}(P^n, i, q)) + \sum_j q_j V(j, (P^n)^T \pi^n) \right\} \qquad (23)$$

Superficially, this resembles the Bellman optimality equation for the value function in a single-agent stochastic MDP, where $s$ is a state, $a$ is an action, $R$ is an immediate reward, and $P(s'|s, a)$ is the probability of transition to state $s'$ from state $s$, given action $a$:

$$V^*(s) = \max_a \{ R(s, a) + \sum_{s'} P(s'|s, a) V^*(s') \} \qquad (24)$$

In equation 23, $q_j$ can be interpreted as a transition probability, conditioned on the fact that the current topic is $i$. The action $q$ selected in the state $(i, \pi^n)$ induces a stochastic transition to a next topic $j$, but the next distribution $\pi^{n+1}$ is given by the deterministic forward equation $\pi^{n+1} = (P^n)^T \pi^n$, where $P^n$ is the true Nash maximizer matrix. This means that $q_j$ does not completely specify the next state $(j, \pi^{n+1})$, and there is a formal difference between $P(s'|s, a) V^*(s')$ and $q_j V(j, (P^n)^T \pi^n)$. Also notice that the Bellman equation sums over all possible next states $s'$, but equation 23 only sums over topics $j$ rather than full states $(j, \pi)$.

### G.2  MULTI-AGENT MDP

Short of modeling every single agent in the MFG, an exact reduction from the MFG to a multi-agent MDP (i.e. Markov game) is not possible. A discrete state space discrete action space multi-agent MDP is defined by $d$ agents moving within a set $S$ of environment states; a collection $\{A_1, \ldots, A_d\}$ of action spaces; a transition function $P(s'|s, a_1, \ldots, a_d)$ giving the probability of the environment transitioning from current state $s$ to next state $s'$, given that agents choose actions $\bar{a} := (a_1, \ldots, a_d)$; a collection of reward functions $\{R_i(s, a_1, \ldots, a_d)\}_i$; and a discount factor $\gamma$.

Let the set of $\pi^n$ (with appropriate discretization) be the state space and limit the set of actions to some discretization of the simplex. The alternative to modeling individual MFG agents is to consider each topic as a single "agent". Now, the agent representing topic $i$ is no longer identified with the set of people who selected topic $i$: topics have fixed labels for all time, so an agent can only accumulate reward for a single topic, whereas people in the MFG can move among topics. Therefore, the value function for agent $i$ in a Markov game is defined only in terms of itself, never depending on the value function of agents $j \neq i$:

$$V_i^\mu(s) = \sum_{\bar{a}} \prod_j \mu_j(\bar{a}_j|s) \left( R_i(s, \bar{a}) + \gamma \sum_{s'} P(s'|s, \bar{a}) V_i^\mu(s') \right) \qquad (25)$$

where $\mu := (\mu_1, \ldots, \mu_d)$ is a set of stationary policies of all agents. However, recall that the MFG equation for $V_i^n$ explicitly depends on $V_j^{n+1}$ of all topics $j$, which would require a different form such as the following:

$$V_i^\mu(\pi^n) = \sum_{P \in \mathbb{S}(\mathcal{G})} \prod_{j=1}^d \mu_j(P_j|\pi^n) \left( \sum_k P_{ik} r_{ik}(\pi^n, P) + \sum_k P_{ik} V_k^\mu(P^T \pi^n) \right) \qquad (26)$$

where the last terms sums over value functions $V_k^\mu$ for all topics $k$. This mixing between value functions prevents a reduction from the MFG to a standard Markov game.

