# OpenReview forum: "Learning Deep Mean Field Games for Modeling Large Population Behavior"
_ICLR.cc/2018/Conference — Accept (Oral)_

### Official Review · AnonReviewer3 · 2017-11-27
**The paper relates the theories of Mean Field Games and Reinforcement Learning  within the classic context of Markov Decision Processes. The method suggested uses inverse RL to learn both the reward function and the forward dynamics of the MFG from data, and its effectiveness is demonstrated on social media data.**

**Rating:** 8
**Confidence:** 3

**Review:**

The paper considers the problem of representing and learning the behavior of a large population of agents, in an attempt to construct an effective predictive model of the behavior. The main concern is with large populations where it is not possible to represent each agent individually, hence the need to use a population level description.  The main contribution of the paper is in relating the theories of Mean Field Games (MFG) and Reinforcement Learning (RL) within the classic context of Markov Decision Processes (MDPs). The method suggested uses inverse RL to learn both the reward function and the forward dynamics of the MFG from data, and its effectiveness is demonstrated on social media data.
The paper contributes along three lines, covering theory, algorithm and experiment.  The theoretical contribution begins by transforming a continuous time MFG formulation to a discrete time formulation (proposition 1), and then relates the MFG to an associated MDP problem. The first contribution seems rather straightforward and appears to have been done previously, while the second is interesting, yet simple to prove. However, Theorem 2 sets the stage for an algorithm developed in section 4 of the paper that suggests an RL solution to the MFG problem. The key insight here is that solving an optimization problem on an MDP of a single agent is equivalent to solving the inference problem of the (population-level) MFG. Practically, this leads to learning a reward function from demonstrations using a maximum likelihood approach, where the reward is represented using a deep neural network, and the policy is learned through an actor-critic algorithm, based on gradient descent with respect to the policy parameters. The algorithm provides an improvement over previous approaches limited to toy problems with artificially created reward functions. Finally, the approach is demonstrated on real-world social data with the aim of recovering the reward function and predicting the future trajectory. The results compare favorably with two baselines, vector auto-regression and recurrent neural networks.
I have found the paper to be interesting, and, although I am not an expert in MFGs, novel and well-articulated. Moreover, it appears to hold promise for modeling social media in general. I would appreciate clarification on several issues which would improve the presentability of the results.
1)	The authors discuss on p. 6 variance reduction techniques. I would appreciate a more complete description or, at least, a more precise reference than to a complete paper.
2)	The experimental results use state that “Although the set of topics differ semantically each day, indexing topics in order of decreasing initial popularity suffices for identifying the topic sets across all days.” This statement is unclear to me and I would appreciate a more detailed explanation.
3)	The authors make the following statement: “ … learning the MFG model required only the initial population distribution of each day in the training set, while VAR and RNN used the distributions over all hours of each day.” Please clarify the distinction and between the algorithms here. In general, details are missing about how the VAR and RNN were run.
4)	The approach uses expert demonstration (line 7 in Algorithm 1). It was not clear to me how this is done in the experiment.

---

> ### Author Response · Authors · 2017-12-07
> **We appreciate the precise summary, and we address each of the questions raised in the review.**
>
> Thank you for highlighting the main points of the paper in detail, and identifying that our contribution lies at the relatively underexplored intersection of RL and models of population behavior. To address the questions raised:
>
> 1. There is a substantial amount of prior work on variance reduction in gradient-based methods for MDPs (Sutton & Barto 1998, Weaver & Tao 2001, Greensmith et al. 2004, Lawrence et al. 2003). Using the policy gradient theorem (e.g. section 13.2 in Sutton & Barto), one can see that subtracting any arbitrary state-dependent function from the action-value estimate does not introduce bias. As can be seen in eq 1 of Greensmith et al., subtracting a baseline reduces variance as long as the covariance is large. Therefore, an optimal baseline can be found to minimize variance (Weaver & Tao 2001). Sutton & Barto 1998 give an empirical demonstration in Figure 2.5 that an appropriately chosen baseline can speed up convergence. We can include a citation to section 3 of Sutton et al. 1999, as it gives an equivalent statement of using the value function as a baseline.
>
> 2. Here is an example: suppose there are three topics (t1, t2, t3), and the initial count of participants at 9am of day 1 is (10, 5, 15). So we reorder the topics to be (t3, t1, t2) and relabel them as (s1, s2, s3). This is what we meant by ``indexing topics in order of decreasing initial popularity.'' On day 2, the topics may be semantically different, e.g. (t4, t5, t6), with initial participation counts (5, 15, 10), so we reorder them to be (t5, t6, t4), and again assign labels (s1, s2, s3). So now both t3 of day 1 and t5 of day 2 are relabeled (i.e. ``identified'') as s1. This is what we meant by ``identifying topic sets across all days.'' This is how we abstract away semantic content and only work with the distribution (i.e. ranking). This reordering lets us interpret our collected demonstration trajectories in a consistent manner, e.g. each trajectory is like running one episode of the constructed MDP, with different starting states (i.e. different initial pi^0) but with the same fixed topic set. Since real populations are influenced by both ranking and semantics, we acknowledge that this method limited the scope of the current work. It suggests a possible extension, e.g. augmenting our basic MFG model to account for topic semantics.
>
> 3. This can be understood from line 4 of Algorithm 2 in Appendix B. For each training episode of the forward RL, we randomly pick a starting pi^0 from the collection of all measured initial distributions. During the single episode, the forward RL never uses the measured pi^1,...,pi^{N-1} because our constructed MDP provides the transition equation, which produces next states pi^{n+1} from pi^n and the action P^n produced by the policy being learned (lines 6 and 7 of Algorithm 2). In contrast, both VAR and RNN are classic examples of supervised learning, where each pi^n,...,pi^{n-m} (for some m) in the training set is used to predict pi^{n+1}. We will supplement Appendix D to describe VAR and RNN in more detail.
>
> 4. Expert demonstration trajectories, sampled from the full set of measured trajectories in line 7 of Alg 1, are used to compute the loss in Equation 13. We take all the state-action pairs of the demonstration trajectories, pass them as a batch into the reward neural network, and add the resulting scalars to get the first term of Equation 13. Learning of the reward is done via gradient descent on this loss, with respect to the neural net parameters W. The process is the same for the second term, which uses trajectories generated from the policy at that iteration.

---

### Official Review · AnonReviewer2 · 2017-11-27
**The technical contributions are good, but the scientific claims should be revised to avoid making unjustified controversial claims**

**Rating:** 8
**Confidence:** 4

**Review:**

This paper attacks an important problems with an interesting and promising methodology.  The authors deal with inference in models of collective behavior, specifically at how to infer the parameters of a mean field game representation of collective behavior. The technique the authors innovate is to specify a mean field game as a model, and then use inverse reinforcement learning to learn the reward functions of agents in the mean field game.

This work has many virtues, and could be an impactful piece. There is still minimal work at the intersection of machine learning and collective behavior, and this paper could help to stimulate the growth of that intersection.  The application to collective behavior could be an interesting novel application to many in machine learning, and conversely the inference techniques that are innovated should be novel to many researchers in collective behavior.

At the same time, the scientific content of the work has critical conceptual flaws.  Most fundamentally, the authors appear to implicitly center their work around highly controversial claims about the ontological status of group optimization, without the careful justification necessary to make this kind of argument.  In addition to that, the authors appear to implicitly assume that utility function inference can be used for causal inference.

That is, there are two distinct mistakes the authors make in their scientific claims:
1) The authors write as if mean field games represent population optimization (Mean field games are not about what a _group_ optimizes; they are about what _individuals_ optimize, and this individual optimization leads to certain patterns in collective behaviors)
2) The authors write as if utility/reward function inference alone can provide causal understanding of collective or individual behavior

1 -

I should say that I am highly sympathetic to the claim that many types of collective behavior can be viewed as optimizing some kind of objective function.  However, this claim is far from mainstream, and is in fact highly contested.  For instance, many prominent pieces of work in the study of collective behavior have highlighted its irrational aspects, from the madness of crowds to herding in financial markets.

Since it is so fringe to attribute causal agency to groups, let alone optimal agency, in the remainder of my review I will give the authors the benefit of the doubt and assume when they say things like "population behavior may be optimal", they mean "the behavior of individuals within a population may be optimal".  If the authors do mean to say this, they should be more careful about their language use in this regard (individuals are the actors, not populations).  If the authors do indeed mean to attribute causal agency to groups (as suggested in their MDP representation), they will run into all the criticisms I would have about an individual-level analysis and more.  Suffice it to say, mean field games themselves don't make claims about aggregate-level optimization.  A Nash equilibrium achieves a balance between individual-level reward functions. These reward functions are only interpretable at the individual level.  There is no objective function the group itself in aggregate is optimizing in mean field games.  For instance, even though the mean field game model of the Mexican wave produces wave solutions, the model is premised on people having individual utility functions that lead to emergent wave behavior.  The model does not have the representational capacity to explain that people actually intend to create the emergent behavior of a wave (even though in this case they do).  Furthermore, the fact that mean field games aggregate to a single-agent MDP does not imply that that the group can rightfully be thought of as an agent optimizing the reward function, because there is an exact correspondence between the rewards of the individual agents in the MFG and of the aggregate agent in the MDP by construction.

2 -

The authors also claim that their inference methods can help explain why people choose to talk about certain topics. As far as the extent to which utility / reward function inference can provide causal explanations of individual (or collective) behavior, the argument that is invariably brought against a claim of optimization is that almost any behavior can be explained as optimal post-hoc with enough degrees of freedom in the utiliy function of the behavioral model. Since optimization frameworks are so flexible, they have little explanatory power and are hard to falsify.  In fact, there is literally no way that the modeling framework of the authors even affords the possibility that individual/collective behavior is not optimal.  Optimality is taken as an assumption that allows the authors to infer what reward function is being optimized.

The authors state that the reward function they infer helps to interpret collective behavior because it reveals what people are optimizing.  However, the reward function actually discovered is not interpretable at all. It is simply a summary of the statistical properties of changes in popularity of the topics of conversation in the Twitter data the authors' study. To quote the authors' insights: "The learned reward function reveals that a real social media population favors states characterized by a highly non-uniform distribution with negative mass gradient in decreasing order of topic popularity, as well as transitions that increase this distribution imbalance."  The authors might as well have simply visualized the topic popularities and changes in popularities to arrive at such an insight. To take the authors claims literally, we would say that people have an intrinsic preference for everyone to arbitrarily be talking about the same thing, regardless of the content or relevance of that topic.  To draw an analogy, this is like observing that on some days everybody on the street is carrying open umbrellas and on other days not, and inferring that the people on the street have a preference for everyone having their umbrellas open together (and the model would then predict that if one person opens an umbrella on a sunny day, everybody else will too).

To the authors credit, they do make a brief attempt to present empirical evidence for their optimization view, stating succinctly: "The high prediction accuracy of the learned policy provides evidence that real population behavior can be understood and modeled as the result of an emergent population-level optimization with respect to a reward function." Needless to say, this one-sentence argument for a highly controversial scientific claims falls flat on closer inspection. Setting aside the issues of correlation versus causation, predictive accuracy does not in and of itself provide scientific plausibility. When an n-gram model produces text that is in the style of a particular writer, we do not conclude that the writer must have been composing based on the n-gram's generative mechanism.  Predictive accuracy only provides evidence when combined in the first place with scientific plausibility through other avenues of evidence.

The authors could attempt to address these issues by making what is called an "as-if" argument, but it's not even clear such an argument could work here in general.

With all this in mind, it would be more instructive to show that the inference method the authors introduce could infer the correct utility functions used in standard mean field games, such as modeling traffic congestion and the Mexican wave.

--

All that said, the general approach taken in the authors' work is highly promising, and there are many fruitful directions I would be exicted to see this work taken --- e.g., combining endogenous and exogenous rewards or looking at more complex applications.  As a technical contribution, the paper is wonderful, and I would enthusiastically support acceptance.  The authors simply either need to be much more careful with the scientific claims about collective behavior they make, or limit the scope of the contribution of the paper to be modeling / inference  in the area of collective behavior.  Mean field games are an important class of models in collective behavior, and being able to infer their parameters is a nice step forward purely due to the importance of that class of games.  Identifying where the authors' inference method could be applied to draw valid scientific conclusions about collective behavior could then be an avenue for future work.  Examples of plausible scientific applications might include parameter inference in settings where mean field games are already typically applied in order to improve the fit of those models or to learn about trade-offs people make in their utility functions in those settings.

--

Other minor comments:
- (Introduction) It is not clear at all how the Arab Spring, Black Lives Matter, and fake news are similar --- i.e., whether a single model could provide insight into these highly heterogeneous events --- nor is it clear what end the authors hope to achieve by modeling them --- the ethics of modeling protests in a field crowded with powerful institutional actors is worth carefully considering.
- If I understand correctly, the fact that the authors assume a factored reward function seems limiting. Isn't the major benefit of game theory it's ability to accommodate utility functions that depend on the actions of others?
- The authors state that one of their essential insights is that "solving the optimization problem of a single-agent MDP is equivalent to solving the inference problem of an MFG." This statement feels a bit too cute at the expense of clarity. The authors perform inference via inverse-RL, so it's more clear to say the authors are attempting to use statistical inference to figure out what is being optimized.
- The relationship between MFGs and a single-agent MDP is nice and a fine observation, but not as surprising as the authors frame it as. Any multiagent MDP can be naively represented as a single-agent MDP where the agent has control over the entire population, and we already know that stochastic games are closely related to MDPs.  It's therefore hard to imagine that there woudn't be some sort of correspondence.

---

> ### Author Response · Authors · 2017-12-07
> **We are grateful for the high quality feedback, and we address the two critical points in the review.**
>
> We greatly appreciate your insightful and high quality feedback. We agree that the intersection of machine learning and modeling collective processes deserves more exploration. Overall, we will improve our language when describing population behavior, and interpretation the inference results more carefully. Below, we address the two critical concerns in detail.
>
> 1. Interpretation of MFG as population-level or individual-level optimization
>
> In all instances where we mention actions, decisions and optimality, we meant individuals. Some examples are ``aggregate effect of individual actions'' (abstract) and ``aggregate decisions of all individuals'' (intro). We did not intend to claim that MFG has a reward that the population itself as an agent tries to optimize. We absolutely agree that MFG models a Nash equilibrium arrived from individual choice, which is shown by equation 6 on page 4. To say that a single-agent policy is optimal for the constructed MDP reward is not the same as saying that any individual optimizes for this constructed reward. Our writing is in accord with the former, not the latter. To improve clarity, we will clearly say that individuals only optimize for the MFG reward, and that the optimal policy for the constructed MDP is only a tool for generating population trajectories, without making any claim about the ontological status of group optimization.
>
> On a related note, could you refer us to particular works that highlight irrational aspects of collective behavior?
>
> 2. Use of reward function to understand behavior
>
> If we understood correctly, the concern about falsifiability is the following: given some optimization framework and demonstration data, there always exists a reward function for which the demonstration is optimal, which means that the hypothesis of optimality is vacuous. To take an extreme case, it is well known that inverse problems suffer from degeneracy, e.g. any behavior is optimal with respect to an all-zero reward (Ng & Russell 2000). But among all possible rewards that can be learned from data, many may not allow the forward dynamics to reproduce data similar to the observations. This is partly the reason that we evaluated predictive accuracy of the model, similar to the evaluation of IRL on task completion in robotics (Finn et al. 2016).
>
> Regarding interpretation of the reward, it is true that we could visualize the statistical distribution of population distributions and transition matrices, to see which types are favored. It is uncertain whether this is easier or harder for extracting insight from data. However, we do not fully understand why summarizing statistical properties of data has equal utility as learning a reward: e.g. in a finite-horizon gridworld with positive reward only at one terminal state, merely looking at statistics of expert trajectories reveals nothing about which state-action pair is good. We accept the advice to restrict our interpretation of reward to be within our model’s representational capacity, due to the lack of semantics in the model.
>
> We acknowledge the warning about using predictive accuracy to justify claims about the physical world. Perhaps saying ``modeled’’, rather than ``understood’’, better conveys our intended message that the MFG only a descriptive model so far. We aimed to tackle the question ``What is a good description of population behavior’’, rather than the question ``Why does the population behave this way''.
>
> We agree that recovering a pre-specified reward in a synthetic MFG is useful to show. We chose to focus exclusively on a real experiment because one of our main motivations was to ground MFG research on real observations.
>
> Response to additional comments:
> 1. We chose to motivate the population modeling problem using these events because: social media enabled a much larger population to participate virtually than would have been possible otherwise; each event involves a large population concentrated on the same general topic but differentiated into discrete subtopics; our experiment data comes from a social media population. We can add this clarification to our introduction.
> 2. The use of r_{ij}(pi, P_i) rather than r_{ij}(pi, P) still couples V_i^n to the actions by individuals in other topics, because the choice of P_i partially determines the next state pi^{n+1}, which determines the actions by individuals at other topics j, which in turn affects V_i^n via the summation over j of V_j^{n+1}. This can be seen by unrolling equation (7) and using (3). The dependence is on actions taken by others at the next time step.
> 3. The MaxEnt IRL procedure maximizes a log likelihood, so viewing it as either statistical inference or optimization should both be valid. Since MDP and RL are optimal control frameworks, we labeled it as optimization.
> 4. We wrote with more emphasis in some places of the text because MFG may be less well-known in the community, hoping that a stronger tone may help with clarity.

---

> > ### Comment · AnonReviewer2 · 2018-01-13
> > **Changes during rebuttal period largely addressed my concerns but room for improvement remains**
> >
> >
> > I appreciate the authors responses to the comments in my review.  The paper is much improved with respect to both of the main issues I brought up, with a few minor exceptions that I assume the authors overlooked in their revisions (detailed below).
> >
> > I think the paper title is misleading.  "Learning optimal behavior policy" sounds like the authors will be making a normative claim. Just to throw something out there to illustrate what I personally would find less misleading, I would say something like "Learning Deep Mean Field Games for Prediction and Statistical Description of Large Populations"
> >
> > In the abstract the authors state "We consider the problem of representing a large population’s behavior policy that drives  the  evolution  of  the  population  distribution  over  a  discrete  state  space."
> > - With the word "drives", the authors here are using the causal language that I complained about previously.
> >
> > In this second round of review, I have realized that calling the MDP a "single-agent" MDP is confusing and inaccurate.  It would more aptly be called a centralized policy for a multiagent MDP, since the single-agent actually controls the distribution of actions of the population of agents. Even this wording is unnecessary, though, and I think simply calling it an "MDP" with no qualifier would be perfectly clear.  (The fact that centralized multiagent MDPs can be solved via single-agent optimization is well-known in multiagent systems.) Including the "single-agent" adjective still hints at the group-as-agent frame that I criticized in my original review.
> >
> > "The learned reward is a step towards understanding population behavior from an optimization perspective."
> > - This sentence is certainly true, in the sense that the methods of the authors might some day be applied to models that could help us understand human optimization patters.  However, the claim is a bit deceptive since people may be optimizing something very different from what is represented / inferred by the MFG model.  I still think it is safest to interpret what the MFG is doing as creating a useful description of the statistics of population behavior.  The authors could see https://www.nature.com/articles/nature11486 for an example of the scientific debate relating to the kind of description that their model offers.
> >
> > "though we do not have access to a ground truth reward function"
> > - There may not be a ground truth reward function. People are not necessarily optimizing anything.
> >
> > "To test the usefulness of the reward and MFG model,"
> > - Test the usefulness for what?  Interpretabiltiy is useful too!  Probably the authors mean "To test the usefulness of the MFG model for prediction"
> >
> > In their "insights" section, the authors state: "Under the assumption that measured behavior is optimal for the constructed MDP, the learned reward function favors states with large negative mass gradient in decreasing order of initial topic popularity, and transitions that increase this distribution imbalance."
> > - To argue for interpretability, the authors should provide insights that are interpretable to lay people.  This sentence is hard to parse and should be expanded in simple language that explains what is learned to a non-technical audience.
> >
> > "The model’s high prediction accuracy supports two approximate descriptions of population behavior: an equilibrium of individuals optimizing the MFG reward; equivalently, an optimal trajectory for the constructed MDP"
> > - I don't know what this sentence means. The epistemology of model comparison for scientific inference if far from settled, barely even ever discussed by anyone as far as I know, and I would be hesitant to conclude anything about the science of collective behavior from the authors' results.  I would say that the authors have shown that data-fitted MFGs are useful for statistical description and prediction.
> >
> > I may have missed some other lingering instances of my same two original complaints, so I implore the authors to do a careful read-through with these criticisms in mind.  For this work to be understood and taken seriously be researchers in collective behavior, it is critical to be precise in wording about what can actually be claimed.   I think the framing of fitted MFGs as useful for statistical description and prediction is accurate and would be interesting to researchers in collective behavior.
> >
> > I still encourage the authors to investigate validating their method via inferring reward functions of well-known MFG from simulation traces (not necessarily for this paper, although that would be nice, but especially if the authors ever submit a longer version, e.g.).
> >
> > Examples of irrational crowd behaviror:
> > - LeBon "Extraordinary Popular Delusions & the Madness of Crowds"
> > - Schiller "Irrational Exuberance"

---

> > > ### Author Response · Authors · 2018-01-29
> > > **We appreciate the detailed suggestions for improvement**
> > >
> > > We appreciate your suggestions for further improving the precision of our language, and we understand the importance of doing so for the work to be useful to researchers in collective behavior.
> > >
> > > We agree with most of your suggestions, and we will make all necessary edits for the final version of the paper if accepted:
> > >
> > > 1. Title. Since we present our work as a method for representation and prediction, and the aim was not to argue for or against the existence of group optimization, we will avoid wording that may be construed as such from the title.
> > >
> > > 2. Using causal language. It was our mistake to overlook this in the abstract in the first revision round. We will make a more thorough review of the whole text.
> > >
> > > 3. We used the phrase ''single-agent'' when describing the MDP, to emphasize the shift in viewpoint from MFG to MDP. MFG focused on the transition vector P_i of people in a discrete state i, along with separate values V_i, while the constructed MDP treats the transition matrix P and aggregated value V(pi) as single entities. However, we agree with your remark that the qualifier ''single-agent'' is redundant and not entirely accurate, given the fact (as you pointed out) that cooperative multi-agent games with global rewards can be solved by finding a single centralized control.
> > >
> > > 4. There certainly needs to be more work in ''data-driven MFG'', in order to understand the connection between a learned reward function and the motivations behind real individual actions, if such connection exists. At the expense of being more verbose, we can clarify the quoted statements. By giving a framework for learning a reward from data, we only take an early step to show that there are such questions to be answered.
> > >
> > > 5. Regarding the ''two approximate descriptions", we simply meant to summarize: an MFG is a model of an equilibrium resulting from individual optimization, an MDP has an optimal trajectory, we showed their equivalence in a special case, and this enabled learning a model with high accuracy. We thought that the word ''description'' was already far enough from the language used in physics (e.g. 'momentum _is_ conserved' as opposed to ''the quantity called 'momentum' is conserved in our descriptive model''; of course, they are entirely justified in speaking that way), but we can qualify it further.
> > >
> > > 6. We will make sure to validate on synthetic MFG in future work, to make a stronger case.
> > >
> > > The linked Nature letter is interesting and pertinent. In the case of preferential attachment, there seems to be a choice between two classes of mechanisms that can reproduce observations, one solely based on randomization, the other involving a notion of optimization and agency. In our case, the discussion is centered on how to describe a reduction from a model of equilibrium among individual actions, to a centralized control problem.
> > >
> > > Thank you for the references to irrational crowd behavior. Economics and finance are some of the early motivators for MFG, and we will need to keep this point in mind if we extend to these areas.

---

### Official Review · AnonReviewer1 · 2017-12-03
**Great paper**

**Rating:** 10
**Confidence:** 5

**Review:**

The paper proposes a novel approach on estimating the parameters
of Mean field games (MFG). The key of the method is a reduction of the unknown parameter MFG to an  unknown parameter Markov Decision Process (MDP).

This is an important class of models and I recommend the acceptance of the paper.

I think that the general discussion about the collective behavior application should be more carefully presented and some better examples of applications should be easy to provide.  In addition the authors may want to enrich their literature review and give references to alternative work on unknown MDP estimation methods cf. [1], [2] below.

[1] Burnetas, A. N., & Katehakis, M. N. (1997). Optimal adaptive policies for Markov decision processes. Mathematics of Operations Research, 22(1), 222-255.

[2] Budhiraja, A., Liu, X., & Shwartz, A. (2012). Action time sharing policies for ergodic control of Markov chains. SIAM Journal on Control and Optimization, 50(1), 171-195.

---

> ### Author Response · Authors · 2017-12-07
> **We appreciate the overall feedback, will make the suggested improvements, and would like to clarify reference [2].**
>
> We highly appreciate your support for the merits of MFG models, especially in synthesis with the well-studied framework of MDP. We agree that our discussion of the collective behavior and interpretation of results should be presented more carefully, and we will update our wording to be more precise. For applications, we will further highlight the synthetic experiments in previous MFG research, and suggest the analogous real-world applications.
>
> Thank you for directing us to alternative work in MDPs with unknown parameters.
> 1. Looking at Burnetas & Katehakis (1997), we see a thematic similarity: they consider the case of an unknown transition law in finite state-action spaces, and also extend the analysis to a model where reward has distribution with unknown parameters dependent on states and actions. Likewise, we consider a reward function with unknown parameters to be learned. Although our constructed MDP has a known deterministic transition, we simulate the MDP and learn via RL to handle continuous states and action spaces. We agree that we should reference Burnetas & Katehakis' contribution to this research theme.
> 2. If we understood Budhiraja, Liu & Shwartz (2012) correctly, they construct a class of action time sharing (ATS) policies that give the same long-term costs as a stationary Markov control, and which enable estimation of unknown model parameters (via deviation from optimal control) while maintaining the same cost per unit time. We agree that the problem of fulfilling a secondary objective while optimizing for a given cost (which doesn't necessarily depend on those secondary parameters) is an interesting one, and seems to be novel for RL research. We can see that the framework of simultaneously estimating unknown parameters while optimizing a known cost is related to the inverse RL framework used in our work, i.e. simultaneously learning an unknown cost and finding an optimal policy. Can we confirm that this is a correct understanding of your comment?

---

### Author Response · Authors · 2018-01-04
**List of changes**

The following changes were made to address our reviewer's comments:

Section 1: Introduction
1. Clarified the role of social media as a common factor among the three examples of large population events.
2. Made it clearer that the assumption of optimality is ascribed to individuals, not to a population.
3. Described applications of MFG in more detail.

Section 2: Related work
1. Added two references to earlier work in unknown MDP estimation.

Section 4: Inference of MFG via MDP optimization
1. Fixed notation for expected start value of policy (in paragraph immediately above Eqn 14).
2. Changed citation for use of value function as a variance reduction technique.

Section 5: Experiments
1. Improved wording to make it clear that MFG is a descriptive model with an optimality assumption.
2. Qualified an explanation for observed performance of MFG compared to other unstructured methods.
3. Clarified the two equivalent ways of speaking about the model, from either the MFG or the MDP perspective.

Section 6: Conclusion
1. Improved wording to convey that we work with MFG only as a predictive framework, leaving aside the ontological status of a reward that drives physical processes in population movement.

Appendix D
1. Added more explanation of difference between how MFG and the alternative methods use training data.
2. More description of VAR and RNN.

---

### Decision · Program_Chairs · 2018-01-29
**ICLR 2018 Conference Acceptance Decision**

**Decision:**

Accept (Oral)

**Comment:**

The reviewers are unanimous in finding the work in this paper highly novel and significant.  They have provided detailed discussions to back up this assessment.  The reviewer comments surprisingly included a critique that  "the scientific content of the work has critical conceptual flaws" (!)  However, the author rebuttal persuaded the reviewers that the concerns were largely addressed.